# The Effect of UV-C Stimulation of Potato Tubers and Soaking of Potato Strips in Water on Color and Analyzed Color by CIE L*a*b*

**Zygmunt Sobol, Tomasz Jakubowski ***  **and Piotr Nawara**

Faculty of Production and Power Engineering, University of Agriculture, Balicka 116B, 30-149 Krakow, Poland; zygmunt.sobol@ur.krakow.pl (Z.S.); Piotr.Nawara@ur.krakow.pl (P.N.)
* Correspondence: tomasz.jakubowski@ur.krakow.pl

**Abstract:** The color of French fries is an organoleptic attribute indicative of this product quality and also a reliable indicator of its safety. The darker the product color, the higher its acrylamide concentration. Acrylamide is an organic compound of the amide group showing neurotoxic and potential mutagenic actions in the human body. The content of acrylamide in fried potato products essentially depends on the contents of reducing sugars in intermediates of French fries' production. The present study aimed to investigate the effect of UV-C irradiation and the soaking of potato strips in water on French fries' color. The study was conducted on French fries obtained from tubers of the Innovator variety. The study was performed with the use of a special chamber for UV-C irradiation of biological samples and the CIE L*a*b* model for color analysis. The results of the study demonstrated that UV-C stimulation of potato tubers before processing had a beneficial effect on French fries' color while the blanching of potato strips and soaking in water at a temperature of 40 °C resulted in the production of French fries lighter in color.

**Keywords:** potato tubers; UV-C radiation; color; potato strips

---

## 1. Introduction

It is known that certain physical methods based on the action of electromagnetic waves induce a response from plants. Potatoes (*Solanum tuberosum* L.) have been stimulated by electromagnetic field [1], microwaves [2,3] and ultraviolet radiation [4]. Potato plants responded to the physical stimulus of alternating electromagnetic field by a reduction in storage losses, microwaves modified sprouting while the short-term UV irradiation of seed potatoes accelerated the emergence of seedlings. With regard to storage processes and crop quality after storage, it is important to note that UV-C irradiation significantly reduced tuber infection by *Rhizoctonia solani* Kühn [5]. The color of fried potato products is an important organoleptic characteristic decisive for product quality and positively correlated with the acrylamide content. The darker the color, the higher the concentration of this compound [6–14]. Therefore, French fry color is a reliable indicator of safety of this food product. The color of French fries and the remaining fried potato products, and thus the acrylamide content, is determined by the content of reducing sugars in intermediates for French fries production [15–17]. The extraction of reducing sugars (mostly from the outer layer) is one of the more efficient methods of reducing their content in the intermediates. The extraction involves soaking of the intermediates in water at a temperature below the starch gelatinization temperature or by blanching [16–18]. The CIE L*a*b* model used for color description is increasingly often used for quality analysis of fruits, vegetables, dairy products and potato products, and for a description of their in-storage and in-process changes [18–21]. According to Pytka [19], the L*a*b* scale has an important advantage over other systems used for this purpose

because it has a separate lightness channel and chromatic components which measure the content of one of colors: green, violet-red, blue and yellow. Another advantage of the L*a*b* model is related to its metric character, allowing for distinguishing color difference ΔE*. Pytka [19] reported that an average observer notices a color difference when ΔE* > 2, and when ΔE* > 3.5 perceives it as a distinct difference.

The aim of the present studies was to determine the effect of UV-C stimulation of potato tubers, soaking of potato strips (intermediates in French fries production) in water, type of frying fat and measurement point on color of the final product.

## 2. Materials and Methods

Storage and laboratory experiments were conducted in the period 2017–2018. The studies were carried out on potato tubers of the Innovator variety. The Innovator potato is most often used for French fry production by European and Polish manufacturers [20]. This is an early variety of culinary type B, with regular round-oblong tubers, shallow eyes and a medium starch content of 14.6%. This variety is very resistant to the darkening of crude and cooked flesh and keeps well. Tubers were stored in a cold store in a single layer on the mesh surface. Storage temperature was 10 °C, and air relative humidity was 90%–95%. Intermediates for French fries production were potato strips measuring $10 \times 10$ mm in cross section and 60 mm in length. Strips were cut lengthwise along the longitudinal tuber axis set between the proximal and distal tuber end. Potato strips were soaked in water to extract reducing sugars in the following combinations: (1) 20 °C for 15 min and (2) 40 °C for 20 min. The study comprised also a group of potato strips (3) blanched at 90 °C for 2 min and the group not soaked in water, which was the control group (0). In order to obtain uniform measurement conditions, before the measurement, the strips were immersed in water and immediately dried (in a two-stage process) using a dry paper towel every time. The same procedure was applied after immersion, namely, the strips were dried in two stages immediately thereafter [22–25]. The studies were performed after 3 months of storage. The chamber (0.63 m$^3$; $0.55 \times 0.95 \times 1.2$) for ultraviolet in the C band (UV-C) irradiation of biological materials was used in the experiment. The device was made of special aluminum with a high coefficient of light reflection. The light source was radiator TUV UV-C NBV 15 W—253.7 nm (total energy flux was 4.0 W) and the durability of the radiator was 8000 h. The radiator was equipped with the option regulation of the height above the chamber bottom (range from 0.4 to 1.0 m). Potato tubers, during irradiation, were situated on a flat bottom (with an area of 0.52 m$^2$) [26,27]. Detailed characteristics of the measuring device and the method of UV-C irradiation were described in papers by Jakubowski and Wrona [26], Jakubowski and Pytlowski [4,5]. Potato tuber UV-C irradiation modes were as follows: (1, 3) irradiation of one side of the tuber for 30 min, (2, 4) irradiation on both sides for 15 min each, and (0) control sample which was not irradiated. The stimulation was performed 48 h before cutting potato strips and their immersion in water (1, 2) and before storage (3, 4). French fries were fried in two types of frying fat: (1) coconut oil, (2) refined rapeseed oil. Frying temperature was 170 °C and frying time was 15 min. French fries were fried in one step until the proper sensory parameters of the final product were reached. Potato-to-oil weight ratio was 1:15. Frying time was chosen in a preliminary experiment on the basis of evaluation by the sensory test panelists. After frying, French fries were drained from oil excess in two steps: 1—on a mesh bottom shaking trays, 2—on a paper towel. The color of French fries was assessed using the CIE L*a*b* method based on instrumental color measurement. In the CIE L*a*b* color space, it is possible to define lightness as the parameter L and chromaticity as parameters a and b. Figure 1A presents the color measurement unit composed of a camera Basler aca-4600-10uc connected with a computer via U3-PCIE1XG205 card operated by Pylon 5 software package. Before measurement, camera was calibrated with a control color palette of white and black (Figure 1B) placed on a white background in order to improve color accuracy. Basler aca-4600-10uc camera possesses an Aptina MT9F002 CMOS sensor with a resolution of $4608 \times 3288$ px and pixel size of $1.4 \times 1.4$ µm.

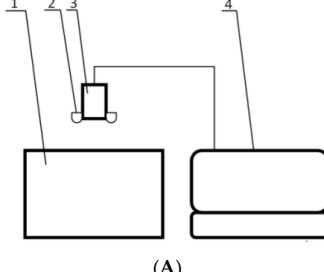
(A)

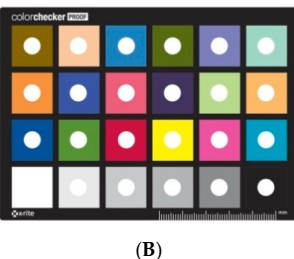
(B)

**Figure 1.** (**A**)—test bed configuration: 1—chamber, 2—lighting, 3—camera, 4—computer; (**B**) reference color palette: X-Rite ColorChecker PRO.

Then, using the graphical environment of LabView2015 software CIE L*a*b* colors were determined at three points of the French fries. French fries during measurement were at an ambient temperature of 20 °C. Each study object was subjected to color analysis on a white background, measurements were taken at three points on each French fry side at three points: (1) at both edges and (2) in the middle. For each experimental combination, measurements were taken on 10 French fries in duplicate. Based on the measurements conducted in the present study, 9600 results, i.e., (L*a* b*) coordinates were obtained. The total color differences were calculated accordingly for the means of experimental combinations.

Data were analyzed according to the following criteria: when absolute color difference (ΔE*) between experimental combinations was below 1, the difference was assumed to be indistinguishable. For ΔE* < 2 the difference was considered to be slight, distinguishable by a person experienced in color intensity analysis, ΔE* from 2–3.5 was defined as medium, distinguishable even by inexperienced person and ΔE* from 3.5–5 was considered to be a distinct difference. ΔE* above 5 was defined as a large color difference.

The obtained results were analyzed using the STATISTICA 13.3 package (parametric analysis of variance in multiple classification) at the assumed significance level of $\alpha = 0.05$. The normality of distribution (Shapiro–Wilk test) and homogeneity of variance (Levene test) were checked.

## 3. Results and Discussion

A repeated measures analysis of variance indicated that all French fries' color parameters established with the use of the CIE L*a*b* model statistically significantly depended on the following factors: stimulation mode, immersion conditions and measurement point. Conversely, the type of frying fat did not statistically significantly influence the parameters under study (Table 1).

**Table 1.** Analysis of variance in multiple classification. Influence of tuber stimulation, immersion conditions of semi-finished products, type of frying and place of color measurement on image brightness (L*), chromaticity of image (a*) and chromaticity of image (b*) determined by CIE L*a*b*.

| Qualitative Predictor | L* | | a* | | b* | |
|---|---|---|---|---|---|---|
| | Value | | | | | |
| | Statistics F Snedecora | Probability of Test | Statistics F Snedecora | Probability of Test | Statistics F Snedecora | Probability of Test |
| Free | 442,735.1 | 0.000000 | 1286.534 | 0.000000 | 29,294.83 | 0.000000 |
| {1} stimulation | 64.7 | 0.000000 | 348.345 | 0.000000 | 342.44 | 0.000000 |
| {2} immersion conditions | 16.0 | 0.000000 | 20.359 | 0.000000 | 68.65 | 0.000000 |
| {3} type of frying | 0.0 | 0.896456 | 2.420 | 0.119807 | 0.59 | 0.443266 |
| {4} place of measurement | 9.9 | 0.001691 | 29.140 | 0.000000 | 14.74 | 0.000125 |

*3.1. The Effect of UV-C Stimulation on French Fries Color Parameters (L*, a*, b*) and Total Color Difference ΔE**

Analysis of French fries' color parameters after UV-C tuber stimulation indicated that lightness (L*) ranged from 85.78 to 90.79 (Table 2). The lowest values were obtained for French fries produced

from tubers stimulated before storage (3, 4) (85.78, 85.93), while the highest values were observed for French fries prepared from tubers stimulated two days before processing (1, 2) (90.15, 90.79). Lightness of control samples (unstimulated tubers) was at the level of 89.58. The a value (color ranging from green to red) assumed values from 0.52 to −8.20. The highest values (0.52, 0.42) were noted for French fries produced from tubers stimulated before storage (3, 4) and the lowest (−7.43, −8.20) for French fries made of tubers stimulated before processing (1, 2) (Table 2). The b value (from blue to yellow) ranged from 32.89 to 55.78. The highest values (55.78, 51.77) were noted for French fries cut from tubers stimulated prior to processing (1, 2) and the lowest (33.06, 32.89) for French fries produced from tubers stimulated prior to storage (3, 4) (Table 2).

Thus, analysis of the color coordinates of the French fries showed that fries produced from tubers stimulated prior to processing were the lightest in color (L* = 90.15, 90.79) with a significant predominance of yellowness (b* = 55.78, 51.77) and slight admixture of greenness (a* = −7.43, −8.20). French fries prepared from tubers stimulated prior to storage were the darkest in color (of all studied samples) (L* = 85.78, 85.93), with the lowest yellowness component (b* = 33.06, 32.89) and a trace of redness (a* = 0.52, 0.42) (Table 2). The colors of French fries are described by the mean values of L*, a*, b*; coordinates for different stimulation modes are visualized in Table 2 as colors generated by Adobe Color CC software. The total color difference ΔE for all experimental combinations of tuber stimulation modes ranged from 0.25 to 24.98. The lowest color difference was observed between stimulation modes 3–4 and was indistinguishable (invisible). The values obtained for the remaining relations indicate that there were distinct or large color differences between the experimental combinations (Table 2).

Analysis of the obtained results can suggest that UV-C tuber stimulation prior to processing resulted probably in the greatest reduction in monosaccharides, and this is why the lightest French fries were obtained for this experimental combination. However, at the present state of research it is not possible to unequivocally resolve whether UV-C irradiation of potato tubers directly stimulates monosaccharide reduction or whether this procedure triggers mechanisms facilitating the leaching of sugars during soaking potato strips in water. The literature does not provide conclusive evidence on the effect of UV-C irradiation on biochemical transformations in potato tubers at different stages of ontogenesis. However, scientific research [8,28] has unequivocally confirmed that UV triggers the photochemical reaction of photoisomerization (and also photosynthesis, photolysis, redox processes) in biological material, which, in potato tubers, can contribute to the transformation of such compounds as flavonoids into their isomers by incident photons. UV-C penetrates into potato tubers across periderm and to directly adjacent flesh (to a depth of ca. 2 mm). Thus, significant changes in biochemical processes caused by UV-C should be sought around the periderm. UV belongs to the most active mutagenic physical agents. This mutagen penetrates into the tissue and dissipates on its molecules, causing their excitation. This phenomenon involves the excitation of outer shell electrons to a higher energy state resulting in the formation of pyrimidine dimmers which alters DNA structure hindering replication process. Flavonoids are polyphenolic ($C_6C_3C_6$) secondary metabolites, containing a flavone carbon backbone and fulfilling a protective role against UV radiation, among other things. They show antioxidant actions and can stop or delay the oxidation of some substances in plants [29–33]. According to Bilger [30] and Havsteen [32] flavonoids are classified based on structural differences, mostly the type of glycosidic bond, number and location of hydroxyl and methoxy groups in the rings, and also on the number of sugar moieties and the presence of sulfonyl groups in sugars or aglycones. In addition, Gulmon and Mooney [33] suggested that under abiotic stress (e.g., UV-C), plants can redirect metabolism to the formation of larger quantities of secondary metabolites (e.g., flavonoids). At this stage of research, it can be assumed that UV-C influences not only formation of flavonoids but also transformations of monosaccharides linked with flavonoids. The results of studies by Wierzbicka et al. [34] on polyphenol contents in eight potato varieties, the tubers contained from 5.21 to 6.69 mg of flavonoids per·100 $g^{-1}$ of fresh weight (represented by: kaempferol, myricetin, quercetin, luteolin). Similar results were obtained by other authors: Ji et al. [35], Ezekiel et al. [36], Albishi [37], Rytel et al. [18]. Interesting results were also obtained from studies by Lin et al. [38] on the effect of UV-C on potato

tuber storage at low temperatures (4 °C). The storage of tubers at low temperatures reduces sprouting and pathogenic infections but simultaneously increases monosaccharide accumulation leading to acrylamide formation during frying process. UV-C irradiation decreased malonaldehyde contents, which indicates that it prevented oxidative damage and reducing sugar accumulation (mostly fructose and glucose). According to some researchers [38], UV-C irradiation regulated gene cascade, sucrose phosphate synthase, invertase inhibitor 1/3 and invertase 1 in potato tubers. That experiment showed that UV-C prevented oxidative damage in tuber cells, which then led to decreased contents of reducing sugars during storage.

### 3.2. The Effect of Soaking Conditions of Potato Strips in Water on French Fries Color Parameters (L*, a*, b*) and Total Color Difference ΔE*

Analysis of parameter L, i.e., the lightness of French fries after soaking of potato strips in water bath showed that its values ranged from 87.21 to 89.72, which indicates high product lightness (Table 3). The highest lightness L* (89.72) was observed for French fries produced from potato strips blanched (3) in water at a temperature of 90 °C for 2 min, while the lowest L* value of 87.21 was seen for French fried prepared from potato strips soaked in water bath at a temperature of 20 °C for 15 min (1) or from untreated strips (0) 88.14. The a values (color ranging from green to red) assumed values from −2.70 to −4.80. The highest value (−2.70) was noted in French fries made of strips unsoaked in water (0) and the lowest (−4.80) when strips were soaked in water at a temperature of 40 °C for 20 min (2). The parameter b values (color from blue to yellow) ranged from 40.38 to 49.38. The lowest vale (40.38) was observed for products prepared from strips soaked in water bath at a temperature of 20 °C for 15 min (1) or unsoaked (0) (41.49).

Analysis of the color coordinates of French fries under study showed that fries produced from blanched strips were the lightest in color (L* = 89.72), with marked predominance of yellowness (b = 49.38) and slight admixture of greenness (a* = −3.16). French fries prepared from strips soaked in water at a temperature of 20 °C for 15 min (1) were the darkest of all studied samples (L* = 87.21), with the lowest contribution of yellowness (b* = 40.38) and a slight touch of greenness (a* = −3.63) (Table 3). The colors of the studied French fries described by the mean values of L*, a*, b* coordinates obtained from the strips soaked in water under different conditions are visualized as colors generated by Adobe Color CC software (Table 3).

The total color difference ΔE* for all experimental combinations with regard to strip soaking conditions ranged from 1.72 to 9.35. The lowest color difference was seen between conditions (1) and (2) and was of borderline distinguishability. A majority of the values obtained for the remaining relations indicated distinct color difference. The fact that French fries from combination (3) showed the lightest bright yellow color with a tint of green probably indicates that the blanching of potato strips was the most efficient method to leach the reducing sugars. Slightly darker French fries were obtained for a combination (2) of strip soaking conditions (Table 3). Similar results were obtained by Pedreschi et al. [16] and Sansano et al. [17] investigating different fried potato products which were fried from strips after extraction of reducing sugars. The mean values of color parameters (L*, a*, b*) and total color difference ΔE* obtained for different conditions of soaking the strips in water.

### 3.3. The Influence of Measurement Point on Color Parameters (L*, a*, b*) of French Fries and Total Color Difference ΔE*

Analysis of French fries color coordinates in relation to the measurement point indicated that French fries were the lightest in color in the middle zone (L* = 88.86) with significant predominance of yellowness (b* = 45.52) and a pinch of greenness (a* = −4.11).

On the edges (1), French fries were slightly darker in color (L* = 88.03), with the lowest amount of yellowness (b* = 43.52) and a tint of greenness (a* = −3.03) (Table 4).

**Table 2.** The average values of the color parameters (L*, a*, b*) and the total color difference ΔE* for the tuber stimulation methods.

| Parameters | Types of Stimulation | Mean Values of Parameters | Relationships of Parameters from Stimulation Modes | Types of Stimulation | Generated Product Colors for Average Values | Relationships | Total Color Difference ΔE* |
|---|---|---|---|---|---|---|---|
| L* | 0 | 89.58 |  | 0 | | | |
| | 1 | 90.79 | | | | | |
| | 2 | 90.15 | | | | 0–1 | 8.45 |
| | 3 | 85.93 | | 1 | | 0–2 | 5.06 |
| | 4 | 85.78 | | | | 0–3 | 17.03 |
| a* | 0 | −3.17 |  | | | 0–4 | 16.88 |
| | 1 | −8.20 | | 2 | | 1–2 | 4.14 |
| | 2 | −7.43 | | | | 1–3 | 24.98 |
| | 3 | 0.52 | | | | 1–4 | 24.82 |
| | 4 | 0.42 | | 3 | | 2–3 | 20.92 |
| b* | 0 | 49.11 |  | | | 2–4 | 20.75 |
| | 1 | 55.78 | | | | 3–4 | 0.25 |
| | 2 | 51.77 | | 4 | | | |
| | 3 | 32.89 | | | | | |
| | 4 | 33.06 | | | | | |

**Table 3.** The average values of the color parameters (L*, a*, b*) and the total color difference ΔE* for the immersion conditions of the semi-finished products in water.

| Parameters | Types of Stimulation | Mean Values of Parameters | Relationships of Parameters from Stimulation Modes | Types of Stimulation | Generated Product Colors for Average Values | Relationships | Total color Difference ΔE* |
|---|---|---|---|---|---|---|---|
| L* | 0 | 88.14 | | 0 | | | |
| | 1 | 87.21 |  | | | | |
| | 2 | 88.71 | | | | | |
| | 3 | 89.72 | | | | 0–1 | 1.72 |
| a* | 0 | −2.70 | | 1 | | 0–2 | 5.77 |
| | 1 | −3.63 |  | | | 0–3 | 8.05 |
| | 2 | −4.80 | | | | 1–2 | 6.73 |
| | 3 | −3.16 | | 2 | | 1–3 | 9.35 |
| b* | 0 | 41.49 | | | | 2–3 | 3.19 |
| | 1 | 40.38 |  | | | | |
| | 2 | 46.83 | | 3 | | | |
| | 3 | 49.38 | | | | | |

**Table 4.** The average values of the color parameters (L*, a*, b*) and the total color difference ΔE* for the point (spot) of color measurement on the French fries.

| Parameters | Types of Stimulation | Mean Values of Parameters | Relationships of Parameters from Stimulation Modes | Types of Stimulation | Generated Product Colors for Average Values | Relationships | Total Color Difference ΔE* |
|---|---|---|---|---|---|---|---|
| L* | 1 | 88.03 |  | 1 | | | |
| | 2 | 88.86 | | | | | |
| a* | 1 | −3.03 |  | | | 1–2 | 2.42 |
| | 2 | −4.11 | | | | | |
| b* | 1 | 43.52 |  | 2 | | | |
| | 2 | 45.52 | | | | | |

The colors of French fries under study described by the mean values of L*, a*, b* coordinates for different strip soaking conditions are visualized as colors generated by Adobe Color CC software (Table 4). The total color difference ΔE* between the middle zone and the edges was 2.42 and was classified as the medium difference distinguishable even by an inexperienced person (Table 4).

The color of fries is one of the organoleptic characteristics that affects the overall assessment of the product. In addition, the color is closely correlated with the content of acrylamide in French fries—the darker the color, the higher the content of this compound in the product. The results of these tests allow the development of quick tests for color evaluation of potato fried products, indirectly quick estimation of their acrylamide content. The numerical notation of coordinates of the analyzed color can be used in machines that automatically qualify fried products (e.g., fries, chips) as safe for consumption. The results of this experiment can be a positive answer in solving the problem presented in Commission Regulation (EU) 2017/2158 of 20 November 2017 establishing mitigation [39].

## 4. Conclusions

1.  UV-C stimulation of potato tubers prior to processing had a beneficial effect on French fries' color, whereas, when applied prior to storage, it changed all French fries color parameters, causing a color shift towards a darker range;
2.  Blanching of potato strips and soaking them in water at a temperature of 40 °C resulted in a lightening of French fries' color compared with French fries prepared from strips soaked in water at a temperature of 20 °C or unsoaked;
3.  French fries edges were slightly darker in color compared with the middle part;
4.  There was no statistically significant difference in French fry color between different frying fats.

## 5. Patents

Jakubowski T. Patent: The method and device for increasing the storage life of potato tubers with the participation of radiation UV-C (in polish; Sposób i urządzenie do zwiększania trwałości przechowalniczej bulw ziemniaczanych przy udziale promieniowania UV-C: P.419392, data zgłoszenia 07-11-2016).

Jakubowski T., Sobol Z. Patent: The method for modifying the color of potato products and a device to modify the color of potato products (in polish; Sposób modyfikowania barwy wyrobów z ziemniaków i urządzenie do modyfikowania barwy wyrobów z ziemniaków: P.425887, data zgłoszenia 11-06-2018).

**Author Contributions:** Conceptualization, Z.S. and T.J.; methodology, T.J., Z.S. and P.N.; validation and formal analysis, P.N., T.J. and Z.S.; investigation, resources, and data curation, T.J., Z.S. and P.N.; writing—original draft preparation, writing—review and editing, and visualization, T.J., and P.N. All authors have read and agreed to the published version of the manuscript.

**Funding:** This research received no external funding.

**Conflicts of Interest:** The authors declare no conflict of interest.

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
