# Peer review of "The Effect of UV-C Stimulation of Potato Tubers and Soaking of Potato Strips in Water on Color and Analyzed Color by CIE L*a*b*"

_sustainability, doi:10.3390/su12083487_

Round 1

Reviewer 1 Report

The article entitled “Application of the CIE L*a*b* Method for the Evaluation of the Colour of Fried Products from Potato Tubers Exposed to C Band Ultraviolet Light” presents useful data.

The article needs substantial amendments, as follows:

The reference guidelines are not followed, for instance, Line 35: Cools et al. [5]; and Line 82 (Jakubowski et al. 2012, 2013, 2015).

Table 2: Explain the alphabet a, b, c, d, and e used in table 2.

Discussion of obtained data is missing: Discuss the obtained data with other methods utilized for evaluating the color of French fries and other food products.

Suggest and highlight the set of most effective parameters which can be applied for UV-C treatment for the retention of color, microbiological quality, and highest consumer acceptance of fried products

Author Response

Response to Reviewer 1 Comments

Thanks the Reviewer for comments

Point 1.  The reference guidelines are not followed, for instance, Line 35: Cools et al. [5]; and Line 82 (Jakubowski et al. 2012, 2013, 2015).

We agree - Improved according to the Reviewer's recommendations

Point 2. Table 2: Explain the alphabet a, b, c, d, and e used in table 2.

We agree - Improved according to the Reviewer's recommendations

a, b, c, d and e - groups of homogeneous variables

Point 3. Discussion of obtained data is missing: Discuss the obtained data with other methods utilized for evaluating the color of French fries and other food products.

We did not have a comparative method in our research.

We heave reference colour palette (we compared to the color palette (Fig. 1B)): X-Rite ColorChecker PRO using to photograph and printer to calibrate color. The 5-point scale method and the linear scaling method were used in sensory evaluation of colour. Additionally, colour parameters (L*, a*, b*) of potato paste were measured using a Minolta CR-310 colorimeter.

Of course, we know that there are instrumental methods.

We make plan to introduce comparative methods in subsequent studies.

Another method of assessment is:

- Tomaszewska M. ,Neryng A. „The influence of thermal treatment environment and storage conditions on colour of ready potato products prepared according to cook-chill method include” measured using a Minolta CR-310 colorimeter,

- Kramer table.

Point 4. Suggest and highlight the set of most effective parameters which can be applied for UV-C treatment for the retention of color, microbiological quality, and highest consumer acceptance of fried products.

The research presented in the manuscript should be treated as preliminary.

At this stage of research and based on the results obtained, we are not able (with high accuracy) to indicate a set of the most effective parameters.

Research in this direction will be the subject of our next experiments.

Reviewer 2 Report

Apparently the objective of this article seems to be to study the effect of ultraviolet radiation on the final color of potato chips. I must confess that I have not quite understood the meaning of "Relations between exposure parameters". What is the difference between 0-1, 0-2, 0-3 and 0-4, if I have understood correctly all these samples must be without irradiation treatment (0), 0-1 and 0-2 would be samples taken two days before preparing the semiproducts and therefore they would be replicates while 0-3 and 0-4 would be samples taken before storage. If this reasoning is correct, then in Table 2 samples such as 1-1, 2-1, 2-2, 3-1, 3-2, 3-3, 4-1, 4-2, 4-3 and 4-4 are missing. I am sorry if I have not been able to interpret these parameters well.

In any case, the article only gives data on different color parameters, the relationship of these parameters should be discussed first in relation to the different types of treatments and finally to the final quality of the product.

Author Response

Response to Reviewer 2 Comments

Thanks the Reviewer for comments

Point 1.  What is the difference between 0-1, 0-2, 0-3 and 0-4

Explanation

The UV-C irradiation of potato tubers was varied by adopting the following parameters: (1, 3) – 30 min. Exposure on one side of the tuber, (2, 4) – 15 min. Exposure at two opposite sides of the tuber. The irradiation was performed two days before preparing the semi-products (1, 2) and before storage (3, 4).

0 – control sample (with out irradiation),

1 – 30 min exposure on one side of the tuber, the irradiation was performed two days before preparing the semi-products,

2 – 15 min. exposure at two opposite sides of the tuber, the irradiation was performed two days before preparing the semi-products,

3 – 30 min exposure on one side of the tuber, the irradiation was performed two days before storage,

4 - 15 min. exposure at two opposite sides of the tuber, the irradiation was performed two days before storage,

We also changed the description of the first column in Table 2 (Relations between exposure parameters - Relations between experiment combinations). In our opinion, this will help us better understand the layout of the experiment.

Round 2

Reviewer 2 Report

Thank for the response, I have finally understood table 2. I would suggest to include in material and methods the explanation include in the response to reviewer. I would also suggest to extend the discussion of the results and to explain clerarer how the UV radiation afectes to the final quality of the french fries.

Author Response

Ansver for Reviewer 2

Thank Reviower 2 to for the comments

1) I would suggest to include in material and methods the explanation include in the response to reviewer.

We agree - Improved according to the Reviewer's recommendation

- as below (the information in manuscript is given in brackets):

(1 – 30 min exposure on one side of the tuber, the irradiation was performed two days before preparing the semi-products and 3 – 30 min exposure on one side of the tuber, the irradiation was performed two days before storage),

(2 – 15 min. exposure at two opposite sides of the tuber, the irradiation was performed two days before preparing the semi-products and 4 - 15 min. exposure at two opposite sides of the tuber, the irradiation was performed two days before storage).

Added explanation

0 – control sample (with out irradiation) - this information was in manuscript 

2) I would also suggest to extend the discussion of the results and to explain clerarer how the UV radiation afectes to the final quality of the french fries.

We agree - Improved according to the Reviewer's recommendation

Explanation

The color of fried and baked products made of potato (including French fries) is correlated with the content of reducing sugars. Product darkening occurs due to Maillard reaction (non-enzymatic browning) that occurs between reducing sugars (glucose, fructose and free asparagine). In the case of potato tubers, reducing sugars are considered the main precursor of brown color in fried and baked products. According to Al-Juhaimi et al. [36] UV-C radiation induces a photochemical reaction in the form of photoisomerization, which can contribute to the transformation of flavonoids into their isomers due to the activity of photons. Therefore (putative), the UV-C radiation can influence the reduction of monosaccharides or initiate mechanisms aiding the process of sugars elution by immersion in water (this can affect the color of French fries).

New position of rferences was added:

Al-Juhaimi F.; Kashif Ghafoor; Mehmet Musa O ̈zcan; M. H. A. Jahurul, Elfadil E. Babiker, S. Jinap F. Sahena M. S. Sharifudin I. S. M. Zaidul.. Effect of various food processing and handling methodson preservation of natural antioxidants in fruits and vegetables. Journal of Food Science and Technology -Mysore- 55(3), 2018, DOI: 10.1007/s13197-018-3370-0.